# The Prevalence and Morphology-Wise Demographic Distribution of Ponticulus Posticus on CT Scans—A Retrospective Observational Study

**DOI:** 10.3390/medicina59040650

**Published:** 2023-03-24

**Authors:** Alin Horatiu Nedelcu, Andrada Hutanu, Irina Nedelcu, Simona Partene Vicoleanu, Gabriel Statescu, Liviu Gavril, Ana Maria Haliciu, Manuela Ursaru, Cristina Claudia Tarniceriu

**Affiliations:** 1Department of Morpho-Functional Sciences I, Discipline of Anatomy, Faculty of Medicine, “Grigore T. Popa” University of Medicine and Pharmacy, Universitatii Str 16, 700115 Iasi, Romania; 2Radiology Clinic, Recovery Hospital, 700661 Iasi, Romania; 3Faculty of Medicine, “Grigore T. Popa” University of Medicine and Pharmacy, Universitatii Str 16, 700115 Iasi, Romania; 4ENT Clinic, “St. Spiridon” County Clinical Emergency Hospital, 700111 Iasi, Romania; 5Department of Surgical Science I, “Grigore T. Popa” University of Medicine and Pharmacy, Universitatii Str 16, 700115 Iasi, Romania; 6Radiology Clinic, “St. Spiridon” County Clinical Emergency Hospital, 700111 Iasi, Romania; 7Hematology Clinic, “St. Spiridon” County Clinical Emergency Hospital, 700111 Iasi, Romania

**Keywords:** ponticulus posticus classification, Kimmerle anomaly, arcuate foramen

## Abstract

*Background and Objectives:* The ponticulus posticus (PP) is a bony bridge that emerges from the posterior aspect of the superior articular process, to connect the posterior arch of the atlas. It is often associated with neurological symptoms. The aim of this study was to obtain an insight into this malformation, and prevalence in the North East region of the Romanian population. *Materials and methods:* This anatomical variant was analyzed through an observational and retrospective study which was carried out in St. Spiridon Hospital Iasi. The duration of the study was 10 months and, a number of 487 patients who presented neurological symptoms without cranio-cerebral traumatisms were enrolled and a computed tomography (CT) scan was performed. We proposed a new classification of PP in five types. The prevalence of PP was calculated and Skewness test, ANOVA test with Bonferroni correction, and Student’s *t*-test were used for statistical analysis. *Results*: Among the sample of 487 patients, PP was found in 170 cases (34.90%) in an age group of 8–90 years (mean age = 59.52 years, SD ± 19.94 years). Type I was found in 11.29%, followed by Type II—8.21%, Type III—5.13%, Type IV—5.54%, and Type V—4.72% (*p* = 0.347). It was 19.5%, mirroring the incomplete type, whereas the complete type was reported in 15.40% of cases (*p* = 0.347), the highest prevalence, namely 41.17% was found in the “41 to 60 years” age group, followed by 36.95% in the “21 to 40 years” group (*p* = 0.00148). The mean age was higher in patients with PP Type III (61.16 years, SD ± 19.98), while patients with PP Type V recorded the lowest mean age (56.48 years, SD ± 22.13). The differences between the comparative average ages on types were not statistically significant (*p* = 0.411). The gender and age were not good predictors of PP Type V (AUC < 0.600). *Conclusion:* according to our study, incomplete types of PP were found to be more prevalent as compared to complete types. No difference between males and females was detected. PP is more frequent in adults and young adults than in the elderly population. It is confirmed that gender and age were not good predictors of the bilateral complete type of PP.

## 1. Introduction

Ponticulus posticus is a bony bridge emerging from the posterior aspect of the superior articulating process, surrounding the vertebral artery and suboccipital nerve, to connect the posterior arch of the atlas. This anatomic variant has been called by different names: pons posticus, arcuate foramen, and Kimmerle anomaly, but ponticulus posticus is currently the most used term [1]. Its history dates to the 12th century [2] when this structure was found in human skeletons, while the first writings were provided much later on by Louis Bolk [3], a Dutch anatomist, in 1906. An overall incidence of ponticulus posticus has been reported to 16.7%. Literature reveals a higher rate in females compared with males and this abnormality is also age dependent [4].

The pathology can be divided into two categories: partial and complete [1]. Nevertheless, according to Miki et al. [5], it was classified into three morphological types: full type—a complete bony ring; incomplete type—some portions of the bony ring are missing; and the calcified type. For an improved and clearer connection between the morphological features and clinical manifestations, Nedelcu et al. [6] proposed a new classification of the arcuate foramen, based on anatomical and radiological aspects, as follows:Type I—unilateral incompleteType II—bilateral incompleteType III—unilateral completeType IV—mixed—associating unilateral complete bridge with contralateral incomplete bridgeType V—bilateral complete

The classification in five types is different from the classification proposed by Stubbs in 1992 [7]. Regarding the symptomatology, in an early study done by Chitroda et al. [8], migraines and chronic types of headaches have been noted to be the major manifestations of ponticulus posticus. Vertigo, diplopia, shoulder, or neck pains were recorded among other clinical presentations of this anomaly. These were clearly linked to the compression of the vertebral artery on its way from the foramen transversum of the atlas to the foramen magnum of the skull. Moreover, Tambawala et al. [9] described part of the symptomatology as contributing to the onset of cervicogenic headache, which represents pain originating from the neck area, but with real feeling and radiation towards the patient’s head. The overall prevalence of this type of headache in the study was 6%. Patients may take it for migraine or tension headache, even though the associated symptoms are particularly correlated with cervicogenic headache: reduced motion of the neck, pain on one side of the face or around the eyes, sensitivity to light and noise, nausea, and blurred vision. This cervicogenic form of headache is highly specific to ponticulus posticus, as it only results from structural damage in the neck area, especially regarding vertebras at the top of the spine.

The diagnostic methods include lateral radiographs, computed tomography (CT), and most likely the Cone beam computed tomography (CBCT) because of its low dose of radiation and short exposure time [10]. Knowing this anatomic anomaly, a careful assessment of the preoperative computerized tomography images is helpful to avoid vertebral artery injury during insertion of a first cervical vertebra (C_1_) lateral mass screw to stabilize the cervical spine. Taking this into consideration, when a complete form exists, it can be easily mistaken for a thickened posterior arch and can result in damaging the artery during C_1_ instrumentation [10].

Considering all of the above, our study aimed to investigate the prevalence and morphological features of arcuate foramen in a targeted Romanian population and to correlate them with any possible gender and age predominance.

## 2. Materials and Methods

A retrospective observational study was carried out in St. Spiridon Hospital—the largest hospital from the North-East region of Romania. The duration of the study was 10 months from January 2019 to October 2019, a number of 487 patients that presented neurological symptoms was enrolled, and a computed tomography (CT) scan was performed. The patients with cranio-cerebral traumatisms were excluded. Out of 487 patients, the number of females (247) was approximately equal to males (240). The subjects having a clear visible skull base with ages between 8 and 90 were therefore included in the study. Prior to the examination, patient or parental informed consent was signed by all patients. Our study was approved by the Ethics Committee of the “St. Spiridon” County Clinical Emergency Hospital- nr. 40/30.03.2022. Moreover, our research is based on clinical diagnostic protocol.

The first step was to assign patients to five age groups as follows: “less than 20 years”, “21 to 40 years”, “41 to 60 years”, “61 to 80 years”, and “more than 81 years”. Images were acquired by Siemens Somatom Emotion 16 CT Scanner and were processed with the Radiant DICOM Viewer software. Each digital tomography was being inspected for either the presence or absence of ponticulus posticus, and further evaluated for a specific type, according to the newest classification: type I—unilateral incomplete, type II—bilateral incomplete, type III—unilateral complete, type IV- mixed—associating unilateral complete bridge with contralateral incomplete bridge and, type V- bilateral complete. A complete PP is one continuous bridge that extends from the posterior aspect of the superior articular process (lateral mass) of first cervical vertebra to its posterior arch (anterior aspect of the posterior tubercle). A partial PP is one that does not extend fully from the posterior lateral mass of C_1_ to the posterior tubercle of posterior arch. Ponticulus posticus can possibly be identified incorrectly as a large dorsal arch of the atlas. A normal posterior arch of the atlas thins out laterally and does not curve up cranially, whereas a PP extends cranially [7]. Two authors examined each image for the presence of PP and noted whether any type of PP was present.

The Skewness test (−2 < S < 2) validates the normality of the series of values, it is used when the examined variable has continuous values and was used because the series of values for age was homogeneous. In the calculation of the significant difference between two or more groups, depending on the distribution of the series of values, at the significance threshold of 95%, for the quantitative variables, the following applies: the *t*-Student test, a parametric test that compares the average values recorded in two groups with normal distributions; the F test (ANOVA) used when comparing three or more groups with normal distributions, applying the Bonferroni correction (Bonferroni post hoc), to reduce the error rate when testing multiple hypotheses.

The ROC curve (Receiver Operating Characteristics) is a two-dimensional curve where on the Y axis we have the sensitivity and on the X axis we have the specificity. This curve helps us measure the efficiency of a model, by plotting the specificity/sensitivity balance as a prognostic factor. The larger the area under the curve (the maximum is 1) the better the model.

Area > 0.9—excellent0.9 > Area > 0.8—very good0.8 > Area > 0.7—good0.7 > Area > 0.6—correct (fair)Area < 0.6—the model is rejected

## 3. Results

Our observational retrospective study included CT scans from head and cervical areas of 487 patients, aged 8 to 90 years, of which 240 were males (49.28%) and 247 were females (50.71%). The series of values for age was homogeneous, suggesting that tests of statistical significance can be applied: variations between 8–90 years old range, mean age is 59.52 years ± 19.94; median is 64 years; the result of the Skewness test *p* = −0.442 (Table 1).

The mean age was slightly higher in the female sex (60.95 vs. 58.06 years; *p* = 0.110) (Table 2).

The total prevalence of posticulus posticus was 170 out of 487 patients (34.90%). Male cases were preponderant: 93 out of 240 (38.75%), compared to female cases: 77 out of 247 (31.17%), with the only limitation being that these numbers do not meet the statistical significance. Type I, Type II, and Type V morphological aspects were present in similar proportions in both genders, while Type III and Type IV were predominant in males (Table 3).

Regarding the arcuate foramen distribution in relation to age, the “41 to 60 years” and “61 to 80 years” intervals had the highest raw numbers: 56 and 62 patients, respectively. In fact, the highest prevalence: 41.17% (56 out of 136 patients) was found in the “41 to 60 years” age group, followed by 36.95% (17 out of 46 patients) in the “21 to 40 years” group. These data have met a statistical significance (*p* = 0.00148) (Table 4).

Considering the prevalence based on the morphological classification, Type I (unilateral incomplete) was found to be the most frequent: 11.29% (55 out of 487 patients), followed by Type II (bilateral incomplete) with 8.21% (40 out of 487 patients), Type III (unilateral complete) with 5.13% (25 out of 487 patients), Type IV (mixed) with 5.54% (27 out of 487 patients), and Type V (bilateral complete) with 4.72% (23 out of 487 patients). The study does not meet the statistical significance (*p* = 0.347), and therefore we are not able to extend the results to the general population (Figure 1, Table 4).

The mean age was slightly higher in patients with ponticulus posticus Type III (61.16 years, SD ± 19.98), while patients with ponticulus posticus Type V recorded the lowest mean age (56.48 years, SD ± 22.13), but the differences between the comparative average ages on types were not statistically significant (*p* = 0.411)-F test (ANOVA) (Table 5).

The Bonferroni correction (Bonferroni post hoc) was applied to reduce the error rate when testing multiple hypothese (Table 6).

By plotting the ROC curve, it is confirmed that gender and age were not good predictors of ponticulus posticus Type V (AUC < 0.600) (Figure 2, Table 7).

Over the course of our study, we have also identified other malformations of the cervical vertebras. Examples of worth mentioning cases were seen in patients with ligament calcifications, including atlanto-occipital ligament, patients carrying dehiscence of the posterior arch, or incomplete atlas ossification in a 6-year-old patient. Another defect was found in a 30-year-old male, associating a dehiscence of the posterior arch of the atlas and in the hyoid bone along with a cleft palate and dental malformations. Moreover, two abnormal fusions were noted: one in first two cervical vertebras, resulting in a cervical synostosis, and the other one observed in an occipitalization of the atlas associated with a C_2_-C_3_ cervical block. These findings might represent the starting point for future studies concerning the cervical spine.

## 4. Discussion

The arcuate foramen was suggested to take part in the posterior atlanto-occipital ligament of the quadrupeds as a supplementary loading on the lateral side. However, it has disappeared over time in bipeds, as the longitudinal extension of superior articular process of first vertebra overtook the role of holding up the skull base [11].

Ponticulus posticus, as an anatomical variant, has a high prevalence in the general population. Despite this, it varies widely in literature reports, between 5.1% and 37.8%, depending on the imagistic method used and the age of the group study [12]. In Elliott and Tanweer [13], a meta-analysis which combined bonny samples and patient studies, the overall prevalence was 16.7%, and an increased incidence of the anomaly in anatomic studies was noticed. The complete type (9.3%) was found to be more frequent than the incomplete type (8.7%), while the unilateral was more frequent than the bilateral, 7.6% and 5.4%, respectively. No significant gender prevalence was outlined.

On the other hand, the study on lateral digital cephalometric radiographs performed by Tambawala et al. [9] on 500 patients noted the incidence rate of arcuate foramen to be at 15.8% with female predominance (17.9%) over males (13.1%). In terms of symptomatology, the authors have strongly correlated this anomaly with head and neck pain. Another study of Giri et al. [14] evaluated 414 lateral cephalometric radiographs of 13 to 41 year-old Nepalese patients, in which ponticulus posticus was found in 35.7% of cases. The prevalence of complete form was only 4.8%, while the incomplete form was preponderant, for about 30.9%. The gender incidence in females exceeded the males: 38.20% and 31.25%. The authors concluded that, even with the method limitations, the clinicians should use lateral cefalograms as a screening method for the Kimmerle anomaly.

In Bayrakdar’s et al. [15] CBCT study, including 181 Turkish children aged between 8 and 18 years, the ponticulus posticus’ prevalence was of 36.5%, with statistically significant preponderance in males rather than females. Bilateral morphologic variants were placed at higher rates than unilateral types, 37 (56.1%) and 29 (43.9%) patients, respectively.

A similar CBCT study conducted in an Italian population and published by Tripodi et al. [16] analyzed images of 524 patients aged between 7–17 years, and indicated that the prevalence of arcuate foramen was 28.24%. The gender prevalence was statistically significant accounting for 35.27% in males and 21.42% in females. Presence of bilateral anomaly exceeded the unilateral type by more than double: 19.08% to 7.25%. The most frequent symptoms correlated with this anomaly were identified to be headaches, tinnitus, and migraines for about 32.43%, 6.08%, and 3.38%, respectively. As for one of their conclusions from the study, ponticulus posticus was shown as a frequent anomaly that needs consideration in case of children’s craniofacial pain.

Nonetheless, there remain numerous studies with contradictory results in the literature. The causes are multiple, starting from the subjectivity of the interpretation and going further to the limits of the method used. From the latter point of view, cephalometric radiography has the most limitations, both in terms of x-ray quality and the possibilities of image processing [16,17]. Along with CBCT, the CT scan reduces most of the limitations. In this respect, our observational retrospective study benefited from a high accuracy imagistic method on a large group of patients. We have identified ponticulus posticus in 170 patients out of 487 (34.90%), with a non-statistically significant preponderance in males (38.75%) compared to females (31.17%). These results are in line with previous CBCT studies, but divergent to most lateral cephalometric studies. For morphological and clinical reasons, we have used a new original classification consisting of five types, which took into consideration the morphological aspects of ponticulus posticus: complete/incomplete and unilateral/bilateral. The most frequent findings have been represented by type I—unilateral incomplete (55 out of 487), and type II—bilateral incomplete (40 out of 487). The complete form incorporated type III and type V for about 5.13% and 4.72%, respectively, along with the mixed type IV, covering 5.54% of the entire study group. In our opinion, this new classification into five types is clinically oriented and more objective than the classification proposed by Stubbs [7] in 1992. The old one has a high extent of subjectivism due to its lack of measurement landmarks. However, one classification does not exclude the other, instead they can even coexist.

As regards the age-wise prevalence, unexpected results were obtained in our study, demonstrating that ponticulus posticus is more frequent in adults from 41 to 60 years and young adults (21 to 40 years) than in the elderly population. Although the calcification of the bony bridge has been indicated as a gradual process over time, we found no clear connection with reference to aging. In order to further validate this conclusion, there is evidence for the “less than 20 years” group having similar prevalence compared to the other groups and even the morphological distribution being similar in all groups with predominance of incomplete forms (Type I and Type II). These observations led us to accept the genetic origin of ponticulus posticus more than the calcification theory. The mean age was slightly higher in patients with ponticulus posticus Type III while patients with ponticulus posticus Type V recorded the lowest mean age. Our study confirmed that gender and age were not good predictors of the bilateral complete type of PP that is the most severe type. The strengths of our study are that we utilized a useful classification with a wide scope, and we used 3DCT which can be considered the gold standard for the diagnosis of this variation. Similar studies have been performed previously using 3DCT. This study does not introduce a new idea but gives information about a certain population and uses an effective classification. A recent study published in 2022 [18] showed that 12.6% of the Egyptian population have PP on their atlas vertebrae either bilaterally or unilaterally. In all cases, the PP are significantly narrower than the ipsilateral transverse foramina of atlas and are susceptible to the appearance of the manifestations of vertebrobasilar insufficiency produced by narrowing of the vertebral artery while passing through the arcuate foramen, especially during extreme head and neck rotation and extension. Another study [19] showed that, apparently, it is unrelated to the skeletal malocclusion type or dental anomalies and indicates the congenital hypothesis of PP’s origin already reported in the literature. A narrative review [20] that included 113 articles showed that the PP compresses the third segment of the vertebral artery, the suboccipital nerve, and the venous plexus, consequently contributing to the incidence of neurological pathologies. When a PP is observed or suspected on a lateral radiograph, it is recommended that a computed tomography (CT) scan of a patient who is about to receive a C1 lateral mass screw be performed, which could determine a safe entry point and the right trajectory of the screw insertion [20].

The limitation of the study is that it was performed in a single center and had a low number of cases for prevalence in a population. Further recommendations are to investigate the genetic origin of ponticulus posticus.

## 5. Conclusions

According to our study, incomplete types of PP were found to be more prevalent as compared to complete types in the North Eastern Romanian population. No difference between males and females was detected. As regards the age-wise prevalence, our study concluded that ponticulus posticus is more frequent in adults and young adults than in the elderly population. It is confirmed that gender and age were not good predictors of bilateral complete type of PP. The importance of ponticulus posticus’ diagnosis stands in acknowledging both the probability of total lack of symptoms or, Conversely, inexplicable vertigo, headaches or migraines, letting alone the absence of treatment or prevention aspects of the malformation being discussed. Taking this into consideration, imagistic explorations such as CBCT, CT scans, or even lateral radiographs are essential for diagnostic reasons and are unanimously recognized. In order to increase awareness of this pathology, our study proposed, introduced, and used a standardized imagistic and clinically oriented classification.

## Figures and Tables

**Figure 1 medicina-59-00650-f001:**
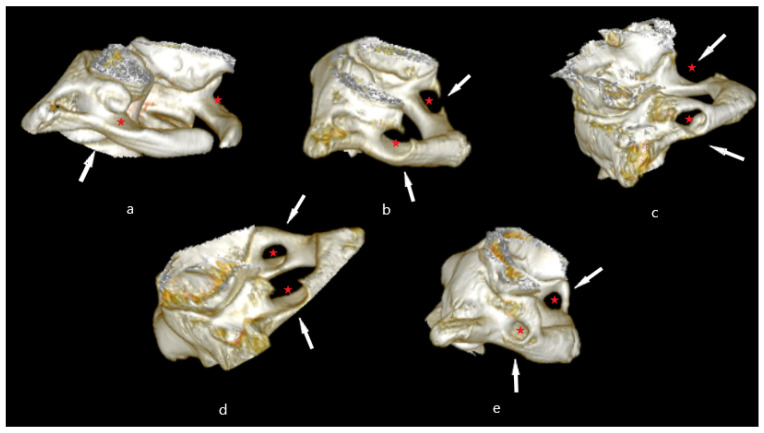
Ponticulus posticus morphologic features: (**a**)—unilateral incomplete (Type I); (**b**)—bilateral incomplete (Type II); (**c**),—unilateral complete right/left (Type III); (**d**)—mixed (Type IV); (**e**)—bilateral complete (Type V). Red star-the place where a bony bridge emerging from the posterior aspect of the superior articulating process, to connect the posterior arch of the atlas.

**Figure 2 medicina-59-00650-f002:**
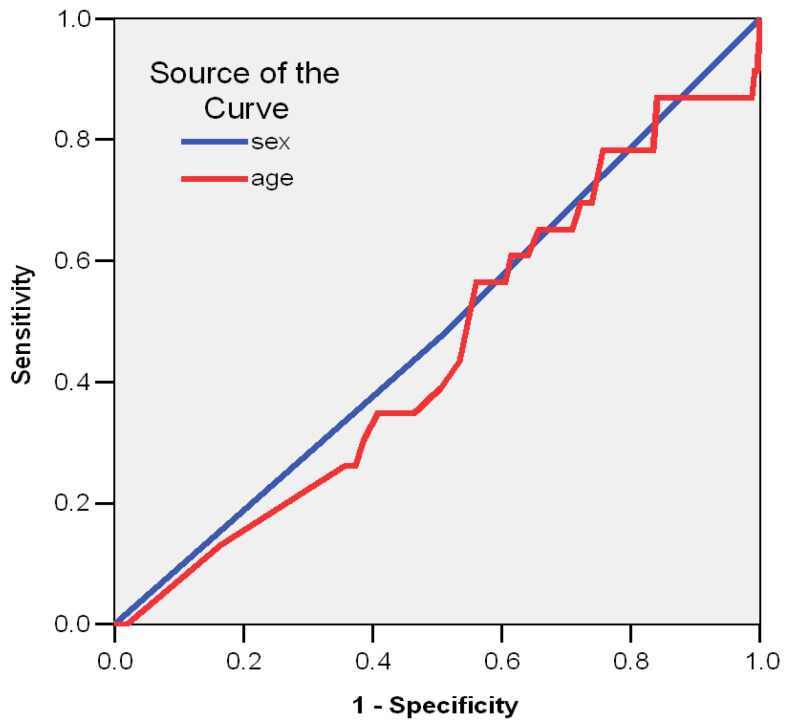
ROC curve that shows the predictors markers for PP type V.

**Table 1 medicina-59-00650-t001:** Descriptive statistics of age.

N	Valid	487
	Missing	0
Mean		59.52
Median		64.00
Std. Deviation		19.94
Variance		33.50
Skewness		−0.422
Std. Error of Skewness		0.111
Minimum		8
Maximum		90
Percentiles	25	48.00
	50	64.00
	75	70.00

**Table 2 medicina-59-00650-t002:** Average age compared by gender.

	N	Mean	Std. Deviation	Std. Error	95% Confidence Interval for Mean		Min	Max	*t*-Student Test
					Lower Bound	Upper Bound			
male	240	58.06	19.731	1.274	55.55	60.57	12	90	
female	247	60.95	20.085	1.278	58.43	63.46	8	90	*p* = 0.110
Total	487	59.52	19.943	0.904	57.75	61.30	8	90	

**Table 3 medicina-59-00650-t003:** Ponticulus posticus—prevalence by gender and morphologic types.

		Type					
Gender	No.	I	II	III	IV	V	Total Prevalence
Female	247	28	18	9	11	11	77 (31.17%)
Male	240	27	22	16	16	12	93 (38.75%)

**Table 4 medicina-59-00650-t004:** Ponticulus posticus—morphology-wise demographic distribution.

		Type					
Age Group	No.	I	II	III	IV	V	Total Age Groups
<20 ani	39	4	3	1	1	3	12 (30.77%)
21–40	46	5	5	3	3	1	17 (36.95%)
41–60	136	14	17	5	13	7	56 (41.17%)
61–80	188	23	13	12	5	9	62 (32.97%)
>80	78	9	2	4	5	3	23 (29.48%)
Total morphological aspects	487	55 (11.29%)	40 (8.21%)	25 (5.13%)	27 (5.54%)	23 (4.72%)	170 (34.90%)

**Table 5 medicina-59-00650-t005:** Average age compared by type of ponticulus posticus.

	N	Mean	Std. Deviation	Std. Error	95% Confidence Interval for Mean		Mini	Max
					Lower Bound	Upper Bound		
don’t have	317	60.34	20.144	1.131	58.11	62.56	12	90
unilateral incomplete	55	60.69	20.195	2.723	55.23	66.15	17	90
bilateral incomplete	40	53.88	16.391	2.592	48.63	59.12	20	89
unilateral complete	25	61.16	19.989	3.998	52.91	69.41	14	89
mixed	27	57.04	19.806	3.812	49.20	64.87	20	89
bilateral complete	23	56.48	22.134	4.615	46.91	66.05	8	89
Total	487	59.52	19.943	0.904	57.75	61.30	8	90

**Table 6 medicina-59-00650-t006:** Multiple Comparisons Bonferroni. Dependent Variable: age.

(I) Type	(J) Type	Mean Difference (I-J)	Std. Error	Sig.	95% Confidence Interval	
					Lower Bound	Upper Bound
don’t have	unilateral incomplete	−0.353	2.913	1.000	−8.95	8.24
	bilateral incomplete	6.463	3.346	0.810	−3.41	16.33
	unilateral complete	−0.822	4.143	1.000	−13.04	11.40
	mixed	3.301	3.998	1.000	−8.49	15.09
	bilateral complete	3.859	4.306	1.000	−8.84	16.56
unilateral incomplete	don’t have	0.353	2.913	1.000	−8.24	8.95
	bilateral incomplete	6.816	4.144	1.000	−5.41	19.04
	unilateral complete	−0.469	4.810	1.000	−14.66	13.72
	mixed	3.654	4.686	1.000	−10.17	17.48
	bilateral complete	4.213	4.952	1.000	−10.40	18.82
bilateral incomplete	don’t have	−6.463	3.346	0.810	−16.33	3.41
	unilateral incomplete	−6.816	4.144	1.000	−19.04	5.41
	unilateral complete	−7.285	5.084	1.000	−22.28	7.71
	mixed	−3.162	4.967	1.000	−17.81	11.49
	bilateral complete	−2.603	5.219	1.000	−18.00	12.79
unilateral complete	don’t have	0.822	4.143	1.000	−11.40	13.04
	unilateral incomplete	0.469	4.810	1.000	−13.72	14.66
	bilateral incomplete	7.285	5.084	1.000	−7.71	22.28
	mixed	4.123	5.535	1.000	−12.21	20.45
	bilateral complete	4.682	5.762	1.000	−12.32	21.68
mixed	don’t have	−3.301	3.998	1.000	−15.09	8.49
	unilateral incomplete	−3.654	4.686	1.000	−17.48	10.17
	bilateral incomplete	3.162	4.967	1.000	−11.49	17.81
	unilateral complete	−4.123	5.535	1.000	−20.45	12.21
	bilateral complete	0.559	5.659	1.000	−16.13	17.25
bilateral complete	don’t have	−3.859	4.306	1.000	−16.56	8.84
	unilateral incomplete	−4.213	4.952	1.000	−18.82	10.40
	bilateral incomplete	2.603	5.219	1.000	−12.79	18.00
	unilateral complete	−4.682	5.762	1.000	−21.68	12.32
	mixed	−0.559	5.659	1.000	−17.25	16.13

**Table 7 medicina-59-00650-t007:** Demographical data for predictor markers of PP type V.

Test Result Variable(s)	Area	Std. Error (a)	Asymptotic Sig. (b)	Asymptotic 95% Confidence Interval	
				Lower Bound	Upper Bound
gender	0.485	0.062	0.806	0.364	0.606
age	0.452	0.061	0.435	0.333	0.570

The test result variable(s): sex, age has at least one tie between the positive actual state group and the negative actual state group. Statistics may be biased. (a) Under the nonparametric assumption. (b) Null hypothesis: true area = 0.5.

## Data Availability

The data presented in this article are available on request from the corresponding author.

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
