# Peer review of "The Prevalence and Morphology-Wise Demographic Distribution of Ponticulus Posticus on CT Scans—A Retrospective Observational Study"

_medicina, 2023, doi:10.3390/medicina59040650_

Round 1

Reviewer 1 Report

Dear authors, Overall, my peer review is a major revision: I suggest revising the manuscript to improve the pitfalls presented. The final goal is to improve the overall clarity of the message to help the reader understand this fundamental topic.

Keywords: use MeSH keywords

Title: Add the type of study.

Abstract:

  1. Prepare the object as per the study title. (The title and objective should be in same order of explanation)
  2. Mention the study design.
  3. Mention the character of the study participants.
  4. Mention the outcome variables measured for the review.
  5. Mention the statistical tests performed for the study.
  6. The results should be presented with 95%CI (upper limit – lower limit) for all the variables with p scores.
  7. The conclusion should be drawn on the basis of the study reports, not on an assumption.

Manuscript

  1. Line 71 – the prevalence rate is already disclosed and discussed in the introduction part.
  2. Mention the acronym of abbreviations when it is used for the first time. (CT, CI, CBCT etc.)
  3. Mention the gaps monitored by the researcher in the previous studies.
  4. Include the clinical significance of this study over clinicians, patients, and researchers after the study hypothesis.
  5. The study duration was four years old. (2019)
  6. Mention clearly the criteria for selection of the study participants.
  7. Mention who has included the study participants in the trial?
  8. Include the ethical approval number.
  9. Include the outcome measures and its reliability and validity.
  10. Include the reference study for sample size calculation.
  11. The samples included in the study is not sufficient enough to generalize the reports.
  12. The statistical tests used for the study was not apt to this study.
  13. Mention where the authors used the ANOVA and student t test?
  14. The results should be presented with 95%CI (upper limit – lower limit) for all the variables.
  15. There is redundant information in the discussion part and not presented in a logical manner. 
  16. The conclusion should be more concise and self-explanatory and drawn on the basis of study reports. 
  17. Add more real-time limitations faced by the researcher and the study. 
  18. Include future recommendations of the study.

Reviewer 2 Report

Dear Editor,

Thank you for giving me the opportunity to review this research.

The study has some strengths and weaknesses:

 The weakness of the study is that it was performed in a single center and had a low number of cases for prevalence in a population.  In addition, image examples containing PP types are not well understood. Instead of showing the whole cranium, it would be appropriate to create a more focused image and mark the variation with an arrow.

The strengths of the study were that they utilized a useful classification with a wide scope and that they used 3DCT which can be considered the gold standard for the diagnosis of this variation. Similar studies have been performed previously using 3DCT. This study does not introduce a new idea. However, It is acceptable in terms of giving information about a certain population and using an effective classification.

I did not notice a problem in terms of study design.

It is recommended to include several more recent (post-2020) anatomical and symptomatological (investigating the relationship of symptoms to structures passing through) studies about arcuate foramen as references.

Kind regards,

Round 2

Reviewer 1 Report

Line 28 – remove the “largest ____________ of Romania.

Keywords: arrange the keywords in a separate line.

Table 5, 6: Use the same type of font
